# Molecular basis of mEAK7-mediated human V-ATPase regulation

Rong Wang [1], Yu Qin[1], Xiao-Song Xie [2] & Xiaochun Li [1,3✉]

The activity of V-ATPase is well-known to be regulated by reversible dissociation of its $V_1$ and $V_o$ domains in response to growth factor stimulation, nutrient sensing, and cellular differentiation. The molecular basis of its regulation by an endogenous modulator without affecting V-ATPase assembly remains unclear. Here, we discover that a lysosome-anchored protein termed (mammalian Enhancer-of-Akt-1-7 (mEAK7)) binds to intact V-ATPase. We determine cryo-EM structure of human mEAK7 in complex with human V-ATPase in native lipid-containing nanodiscs. The structure reveals that the TLDc domain of mEAK7 engages with subunits A, B, and E, while its C-terminal domain binds to subunit D, presumably blocking $V_1$–$V_o$ torque transmission. Our functional studies suggest that mEAK7, which may act as a V-ATPase inhibitor, does not affect the activity of V-ATPase in vitro. However, overexpression of mEAK7 in HCT116 cells that stably express subunit a4 of V-ATPase represses the phosphorylation of ribosomal protein S6. Thus, this finding suggests that mEAK7 potentially links mTOR signaling with V-ATPase activity.

[1] Department of Molecular Genetics, University of Texas Southwestern Medical Center, Dallas, TX 75390, USA. [2] Eugene McDermott Center for Human Growth and Development, University of Texas Southwestern Medical Center, Dallas, TX 75390, USA. [3] Department of Biophysics, University of Texas Southwestern Medical Center, Dallas, TX 75390, USA. ✉email: xiaochun.li@utsouthwestern.edu

 **1**

The vacuolar-type $H^+$-ATPases (V-ATPases), which localize to the plasma membrane, endosomes, and membrane-bound vesicles, are ATP-driven proton pumps which regulate intracellular pH homeostasis[1,2]. It has been shown to play an important role in autophagy[3], nutrient metabolism[4,5], and Wnt, mTOR, and Notch signaling[6–8]. Dysfunction of V-ATPase leads to kidney, neuron, skin, muscle, and bone diseases[9]. V-ATPase is a rotary machine conserved from yeast to mammals. It consists of two domains: the ATP-hydrolytic $V_1$ domain and the proton-translocation $V_o$ domain[10–12]. The $V_1$ domain, containing sixteen subunits ($A_3B_3CDE_3FG_3H$), catalyzes the hydrolysis of ATP by the $A_3B_3$ heterohexamer to trigger the $V_1$–$V_o$ torque transmission via subunits DF. The $V_o$ domain, containing subunits $ac_9c''de$Ac45 and two unique mammalian components, (pro)renin receptor (PRR) and RNaseK[13–15], is responsible for transporting protons from the cytosol to the lumen of organelles or extracellular space.

Several exogenous products have been identified as V-ATPase inhibitors: (1) Bafilomycin, salicylihalamide A, and their derivatives, which are small molecules of the macrolide class, bind to the $V_o$ domain, thereby preventing proton translocation[16,17]; (2) SidK, a *Legionella pneumophila* effector protein[18], binds to the $A_3B_3$ heterohexamer, abolishing ATP hydrolysis[13,15]. The activity of mammalian V-ATPase in vivo is regulated through assembly and disassociation of its $V_1$ and $V_o$ domains by various stimuli, including growth factors stimulation, amino acid starvation, glucose concentration, and cellular differentiation[19,20]. Modulation of V-ATPase trafficking and the expression level of its subunits have also been shown to regulate the V-ATPase activity[21]. Since previous studies have demonstrated the physiological importance of V-ATPase in signaling pathways[6–8], investigations on finding an endogenous V-ATPase modulator may provide molecular insights into V-ATPase activity regulation and V-ATPase-mediated cellular signal transduction.

Here, we discover an endogenous V-ATPase modulator mEAK7 and determine the cryo-EM structure of mEAK7-bound V-ATPase in native lipid-containing nanodiscs. The structural analysis suggests that mEAK7 may function as a V-ATPase inhibitor. Although the interaction of mEAK7 with V-ATPase does not affect the activity of V-ATPase in vitro, overexpression of mEAK7 in HCT116 cells that stably expresses subunit $a4$ of V-ATPase represses the phosphorylation of S6. Thus, this finding suggests that mEAK7 potentially modulates mTOR signaling via binding V-ATPase.

## Results

**Overall structure of mEAK7-bound V-ATPase.** We hypothesized that in different cellular environments (e.g., at a high V-ATPase expression level, during cell death, or under mechanical stimulation), cells may employ an endogenous modulator to regulate V-ATPase activity through direct interaction. However, our previously purified V-ATPase from animal tissue is considered a homogenous source in a stable condition[14,22]. It is technically challenging to capture modulators by changing the local environment of animal tissues rather than cells. Thus, our numerous attempts to find putative V-ATPase modulators have failed.

Recently, a study showed that intact active V-ATPase could be purified from a HEK293 cell line that stably expressed subunit $a4$[23]. We took advantage of this remarkable work to seek for modulators under different cell culture conditions. We generated an Expi293 cell line (derived from the HEK293 cell line, Thermo Fisher) that stably expressed Flag-tagged human subunit $a4$ (Expi-a4 cells). We transfected cDNA encoding the different proteins that potentially regulate V-ATPase into Expi-a4 cells,

then cultured the cells in various conditions and purified the complex using anti-Flag M2 resin. Fortunately, we were able to gain insights into the nature of an endogenous inhibitor through this approach. We transfected aldolase A, which facilitates V-ATPase assembly[24,25], into Expi-a4 cells to increase the population of intact V-ATPase in cells. The V-ATPase was purified with the nanodiscs scaffold protein MSP1E3D1 and reconstituted into native lipid-containing nanodiscs according to an established protocol (Supplementary Fig. 1a)[23].

The resulting complex particles appeared homogenous in electron micrographs after vitrification (Supplementary Fig. 2a and b). During the data processing, an additional electron density was observed in the $V_1$ domain close to the c-ring that we could initially not account for. We subsequently generated a local mask to conduct 3D classification (Supplementary Fig. 2c). A class, containing ~2% of total particles, revealed an unknown protein bound to the $V_1$ domain of V-ATPase (Supplementary Fig. 2c). The overall resolution of the intact V-ATPase is 4.08-Å (Supplementary Table 1). Following our previously established protocol[16], local refinement was used by specific $V_o$ and $V_1$ masks, improving the cryo-EM quality to allow the unambiguous assignment of most residues (Supplementary Figs. 3 and 4). The resulting maps after local refinement were merged by Phenix for model building. The subunits B2, C1, E1, G1, $a4$, $d1$, and $e1$ were built into the final model based on the expression distribution in kidney tissues and a previously published V-ATPase structure from HEK293 cells[15].

To identify the unknown protein bound to the $V_1$ domain, we tentatively built a poly-alanine model into the cryo-EM map assisted by the discernible secondary-structure elements. In parallel, mass spectrometry analysis of our sample identified several candidate proteins (Supplementary Fig. 5). We compared the AlphaFold-predicted structures[26] of these candidates to our preliminarily structural model and found that mEAK7 (mammalian Enhancer-of-Akt-1-7) is the co-purified protein bound to the intact V-ATPase (Fig. 1a, b and Supplementary Fig. 6). mEAK7, which is expressed in many types of cell lines (e.g., HEK cells, lung carcinoma cells and head/neck squamous carcinoma cells), is anchored on the lysosomal surface by its N-terminal myristoyl-glycine[27]. It consists of three domains: the N-terminal domain (NTD, residues 1-229) including nine short α-helices, residues 230-413 presenting a TLDc fold (TLD) with 12 β-strands, which plays a role in the oxidative stress response[28], and the C-terminal domain (CTD, residues 414-456) including two α-helices which potentially interact with mTOR (Fig. 1b)[27]. The TLD and CTD domains of mEAK7 were well determined in the cryo-EM map (Supplementary Fig. 6b and c). In contrast, the linker between TLD and CTD (residues 414-423) was not observed, and the cryo-EM map of the NTD only revealed its secondary structural features but not the side chains owing to the lower local resolution (Supplementary Fig. 6b).

**The interaction details between mEAK7 and V-ATPase.** The three AB heterodimers of the $V_1$ domain are in distinct conformations ($AB_{semi}$, $AB_{closed}$, and $AB_{open}$) (Fig. 1c). Unlike in previously determined V-ATPase structures[13–15], the entire mEAK7-bound complex stays in state 2, while the other two states were not observed (Fig. 1c, d). Structural analysis shows that there are five interfaces between mEAK7 and V-ATPase (Fig. 2): (1) TLD-subunit E (Fig. 2b), (2) TLD domain-subunit $A_{closed}$ (Fig. 2c), (3) TLD and CTD domains-subunit $B_{semi}$ (Fig. 2d), (4) CTD-subunit $A_{semi}$ (Fig. 2e), and (5) CTD-subunit D (Fig. 2e). The NTD of mEAK7 does not contact the V-ATPase, while the TLD is engaged with subunits E, $A_{closed}$, and $B_{semi}$. The interfaces between the TLD and three V-ATPase subunits are

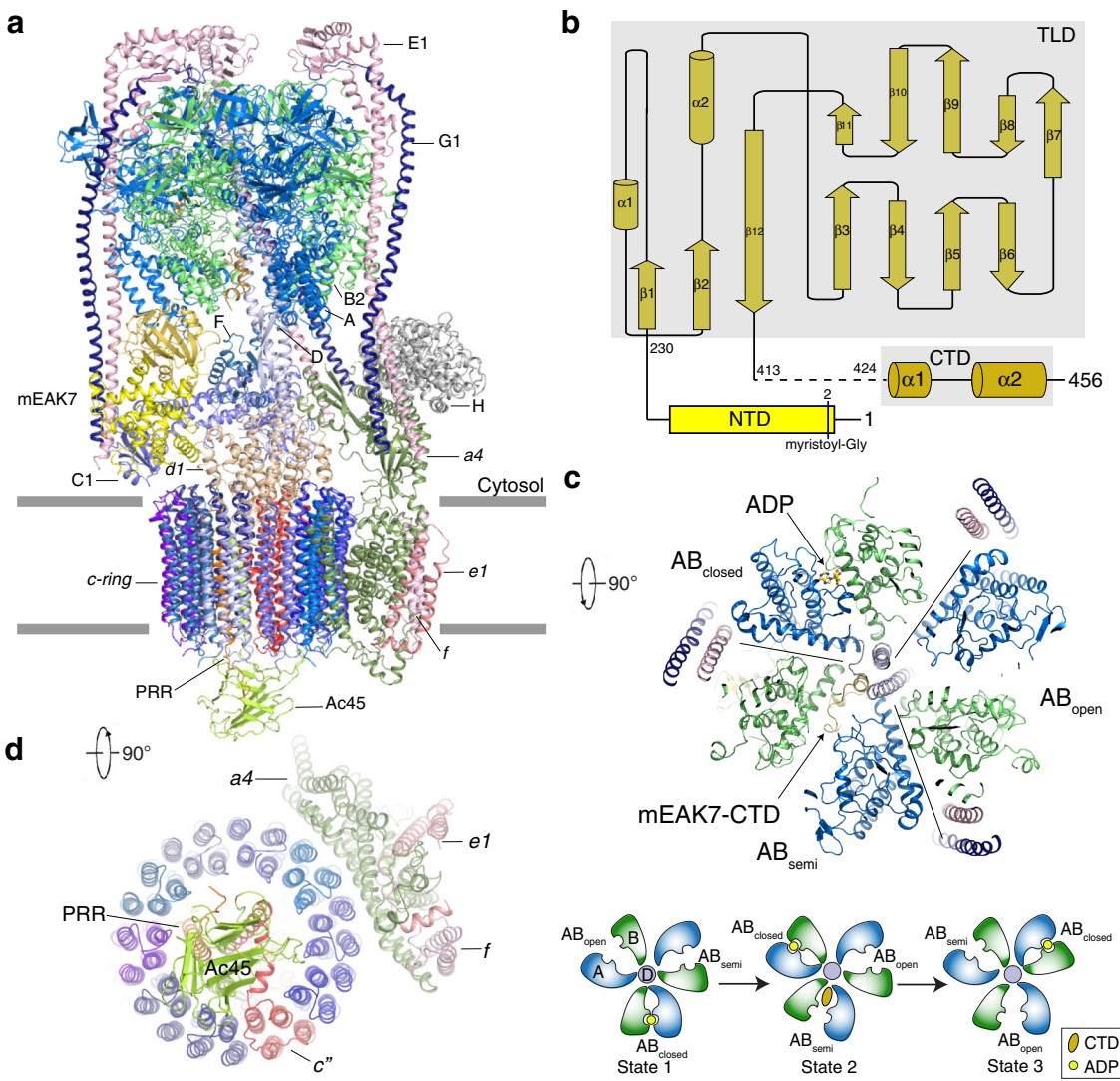

**Fig. 1 Overall structure of mEAK7-bound V-ATPase from an endogenous purification of HEK293 cells. a** Overall structure showing mEAK7-bound V-ATPase viewed from the side of the membrane. **b** The schematic diagram of full-length mEAK7. **c** The cytosolic view of the V-ATPase. A cartoon representation shows the conformations of three AB heterodimers in the distinct states. **d** The luminal view of V-ATPase. Each subunit is colored and indicated.

mainly generated by hydrophilic interactions (Fig. 2b-d). The specific mechanism of how mEAK7 binds to V-ATPase remains unclear, it is possible that the TLD facilitates mEAK7 docking to the $V_1$ domain and may allow the CTD to insert into the cavity in the middle of the $AB_{semi}$ heterodimer (Fig. 2a). The CTD-helix α2 binds subunit D via several salt bridges (Fig. 2e) and hydrophobic contacts between Leu436 and Ile439 of mEAK7 and Ile172 and Ile176 of subunit D (Fig. 2e).

When AB heterodimers hydrolyze ATP, the conformational changes of AB heterodimers trigger the rotation of subunit D and then induce the $V_1 - V_o$ torque transmission, promoting proton translocation through subunit $a^1$. Structural comparison between mEAK7-bound V-ATPase and intact active V-ATPase shows that the CTD-α2 pushes a helix of $B_{semi}$ away from subunit D (Fig. 2f). As a result of this conformational change, the distance between R168-Cα (subunit D) and V438-Cα (subunit $B_{semi}$) increases by 4 Å (Fig. 2f). The extensive interactions between subunit D and the CTD-α2 of mEAK7 prevent the rotation of subunit D and further abolish the torque transmission through contacts among subunits D, $d$, and the c-ring.

**The conformational change of $V_1$ domain**. Although there are three AB heterodimers and three subunits E in the $V_1$ domain, the CTD of mEAK7 only binds to $AB_{semi}$ in V-ATPase according to our structural observations (Fig. 1a). The cavity of $AB_{closed}$ is not large enough to accommodate the CTD (Fig. 3a), while the cavity of $AB_{open}$ is too wide to contact the CTD tightly (Fig. 3b). Therefore, the CTD can only bind to $AB_{semi}$. In state 1 and state 3, subunits H and C would cause steric hindrance to recruit mEAK7 to $AB_{semi}$, respectively (Fig. 3c).

Superimposing subunits EGD and $AB_{semi}$ of the apo bovine V-ATPase structure in state 2 onto mEAK7-bound V-ATPase shows that mEAK7 binding triggers a conformational change involving several helices in $AB_{semi}$ that prevents a steric clash (Fig. 3d). In contrast, the conformations of $AB_{open}$ and $AB_{closed}$ in apo and mEAK7-bound structures are similar. mEAK7 also induces a shift of subunits EG due to the interaction between subunit E and the TLD (Fig. 3e); in contrast, the subunits D in both structures do not reveal a considerable conformational change (Fig. 3f). A comparison with SidK-bound human ATPase structure shows that $B_{semi}$ and subunits EG shift away

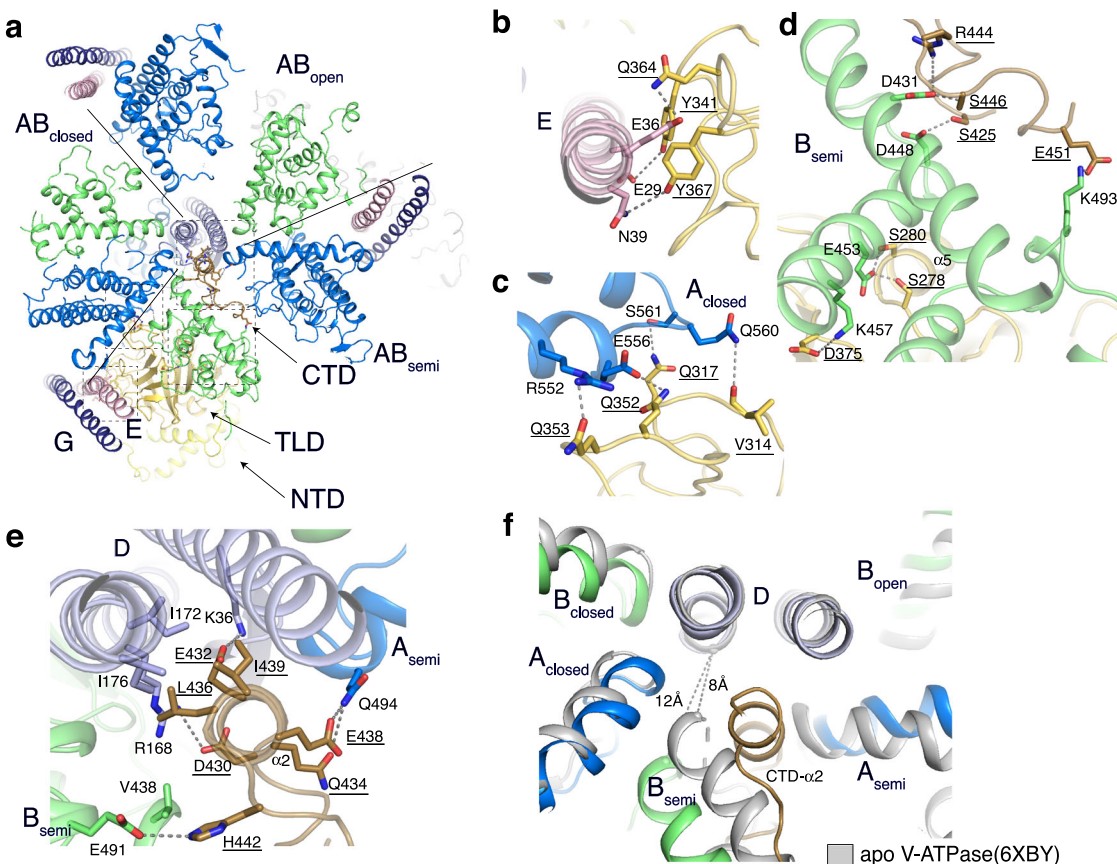

**Fig. 2 Interaction details between mEAK7 and the $V_1$ domain. a** Overall structure showing the interaction between mEAK7 and $V_1$ domain. **b** The interaction between the TLD and subunit E. **c** The interaction between the TLD and subunit $A_{closed}$. **d** The interaction between the TLD, CTD domains and subunit $B_{semi}$. **e** The interaction between the CTD and subunits D and $A_{semi}$. The residues are shown in sticks and the hydrophilic interactions are indicated by dashed lines. The residues of mEAK7 are underlined. **f** Structural comparison with intact active bovine V-ATPase reveals CTD-α2 blocking the contact between subunit D and $B_{semi}$. The distance between Arg168-Cα (subunit D) and Val438-Cα (subunit $B_{semi}$) is indicated by dashed line. The hsV-ATPase is shown with the indicated color while apo-V-ATPase is represented in gray.

from mEAK7 (Fig. 3g, h). Still, there is no apparent structural shift in both $A_{semi}$ and subunit D (Fig. 3i). Structural comparison does not reveal notable conformational changes of the subunit *a* and the c-ring in the mEAK7-bound, SidK-bound, and apo-V-ATPases (Supplementary Fig. 7). This is consistent with that mEAK7 regulates V-ATPase activity via its binding to the $V_1$ but not $V_o$.

The TLDc-domain containing proteins, which play a role in the oxidative stress response and neurological diseases, have been reported to associate with V-ATPase[28–30]. Oxr1 inhibits $V_1$-ATPase activity in yeast[30], which is consistent with our speculation on the role of mEAK7. NCOA7, a human nuclear receptor co-activator, had been shown to bind V-ATPase[31]. The overall structure of NCOA7 shares a similar fold with mEAK7-TLD with an R.M.S.D (root-mean-square deviation) of 0.8 Å (Supplementary Fig. 8). Interestingly, it has been shown that NCOA7 is a V-ATPase activator that promotes cytoplasmic vesicle acidification and lysosomal protease activity[31]. NCOA7 only has the TLD but not the CTD, suggesting that it cannot interfere with $V_1 - V_o$ transmission.

**mEAK7 may act as a V-ATPase inhibitor**. To validate the function of mEAK7 in V-ATPase regulation, we first measured ATPase activity in the presence of mEAK7. Since the yield of endogenous V-ATPase from HEK293 cells is low, we employed the endogenously purified V-ATPase from bovine brain to

perform all in vitro assays. The mEAK7 binding sites are highly conserved in these two species. To confirm that bovine V-ATPase can be a competent substitute for human V-ATPase in these experiments, we purified human mEAK7 from *E. coli* (Supplementary Fig. 1b) and collected the cryo-EM data from a mixture of *homo sapiens* mEAK7 and *bos taurus* V-ATPase (*hs*mEAK7-*bt*V-ATPase complex) at 10:1 molar ratio. The structural analysis showed that human mEAK7 could form a complex with bovine V-ATPase similar to human V-ATPase (Supplementary Fig. 9 and 10). We then employed the human mEAK7 and mEAK7 CTD deletion (mEAK7del) and mixed V-ATPase from bovine brain at 20:1 and 100:1 (mEAK7: V-ATPase) molar ratios. Both mEAK7 and mEAK7del did not demonstrate notable effects on the ATPase activity compared to the treatment with bafilomycin A1 (Fig. 4a). ATP-driven proton translocation assay was also performed to assess the function of mEAK7. In this assay the endogenous V-ATPase purified from bovine brain was reconstituted into proteliposomes, incubated with mEAK7 or mEAK7del at a 20:1 (mEAK7: V-ATPase) molar ratio and mEAK7 functionality was determined by the quenching of acridine orange fluorescence. The result of the proton translocation assay was consistent with the ATPase activity assay (Fig. 4b).

As mEAK7 doesn't interfere with the enzyme activity in vitro, a cell-based assay was performed to explore the role of mEAK7. Activation of V-ATPase is associated with the efflux of amino acids from the lysosomal lumen, and macrolide

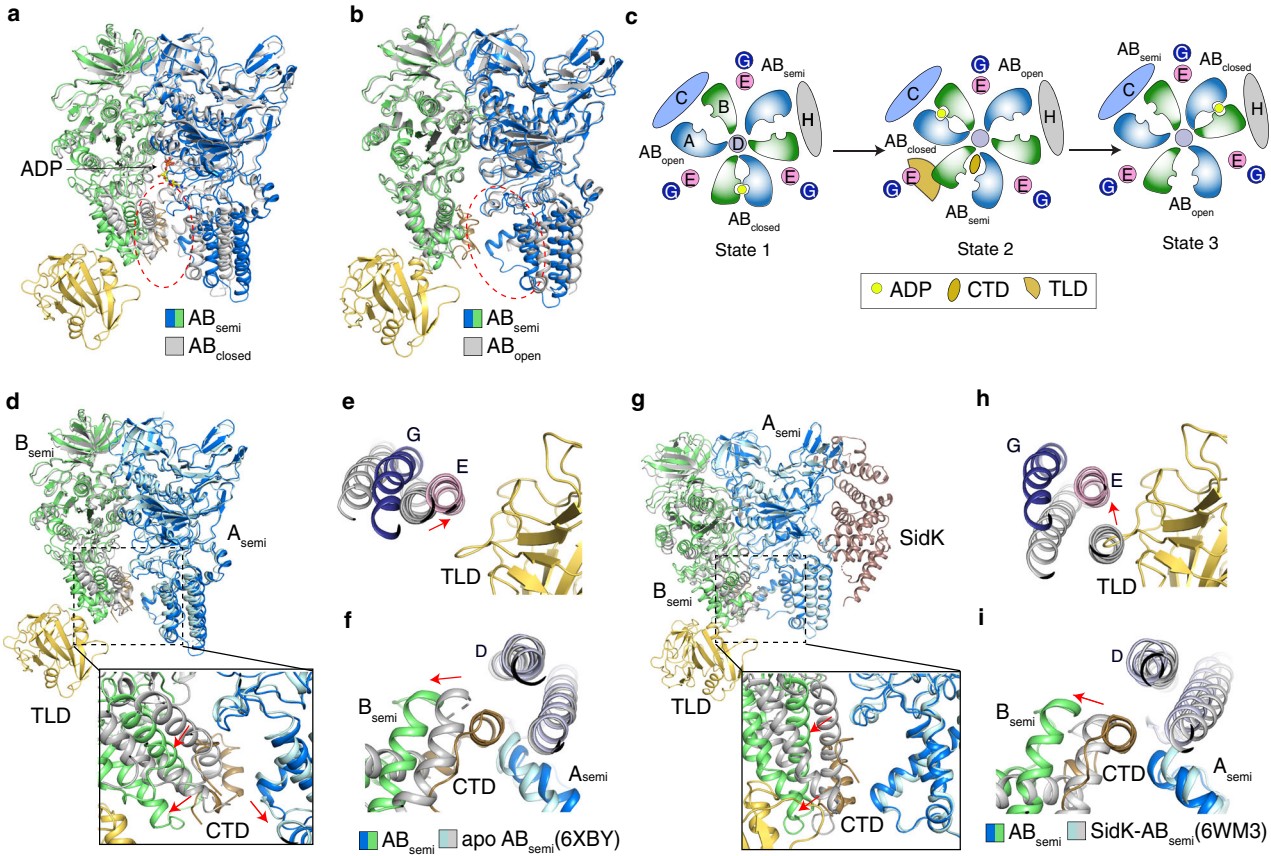

**Fig. 3 Structural comparison reveals the conformational changes of $V_1$ domain owing to the mEAK7 engagement. a** Structural comparison of $AB_{closed}$ to mEAK7 bound $AB_{semi}$. **b** Structural comparison of $AB_{open}$ to mEAK7 bound $AB_{semi}$. **c** The cartoon model explains the uniqueness of the mEAK7 binding site. **d**–**f** The conformational changes of V-ATPase compared to apo bovine V-ATPase. **g**–**i** The conformational changes of V-ATPase compared to SidK-bound human V-ATPase. All conformational changes are indicated by either a red oval or arrows.

V-ATPase inhibitors have been shown to block mTORC1 activation, indicating that V-ATPase activity is indispensable for mTOR signaling[32]. To validate the function of mEAK7 in V-ATPase-mediated mTOR signaling, we generated a monoclonal stable HCT116 cell line that constitutively expresses subunit *a4* (HCT-a4). Immunofluorescence experiment showed that over-expressed *a4* subunit could localize in the lysosome (Supplementary Fig. 11), suggesting the potential role of the *a4* subunit-containing V-ATPase in regulating mTOR signaling through mEAK7. We first transfected cDNA encoding non-tagged wild-type human mEAK7 into HCT-a4 cells. After 36–48 h, the cells were harvested and lysed. Western blotting analysis shows that after bafilomycin A1 treatment, the phosphorylation of S6 ribosomal protein at $Ser^{235/236}$ has been eliminated entirely (Fig. 4c, lines 1 and 2 and Supplementary Fig. 12). With mEAK7 overexpressed HCT-a4 cells, the ($Ser^{235/236}$) p-S6 level also considerably attenuates (Fig. 4c, lines 3–6 and Supplementary Fig. 12), and the ($Ser^{65}$) p-4EBP1 also exhibits weak intensity. Compared to after bafilomycin A1 treatment, the phosphorylation of S6 retained a little activity when mEAK7 is overexpressed (Fig. 4c, lines 2, 5 and 6). We speculate that the interactions between mEAK7 and the $V_1$ domain in cells may be more dynamic or regulated by the cellular environment or other factors causing less inhibitory potency. The phosphorylation of S6 is retained at a similar level in the HCT116 cells. It is possible that HCT-a4 cells might represent a specific class of cells whose V-ATPase level has been upregulated, causing the stronger inhibitory effect by mEAK7. Therefore, this finding

demonstrates that mEAK7 may interfere with mTOR signaling in HCT-a4 cells.

## Discussion

In this manuscript, we discovered that mEAK7, a lysosomal surface-anchored protein, can bind the $V_1$ domain of intact V-ATPase and may function as an inhibitor of V-ATPase. Since V-ATPase expression is upregulated in cancer, V-ATPase inhibitors have been considered for cancer treatment[33]. V-ATPase inhibition through the engineered expression of mEAK7 may delineate another novel approach for human malignancies. Recently, the structure of mEAK7 bound to V-ATPase in the presence of a bacterial V-ATPase inhibitor SidK has been made available online[34]. SidK was used to purify endogenous V-ATPase from the porcine kidney, then recombinant mEAK7 was mixed with the purified SidK-bound V-ATPase in vitro for structure determination. The authors proposed that mEAK7 may be a V-ATPase inhibitor based on their structural analysis. Additionally, the authors found that mEAK7 does not inhibit purified V-ATPase activity and mEAK7 overexpression in cells also does not alter lysosomal or phagosomal pH, which are all consistent with our conclusion. The authors thought the interaction between mEAK7 and V-ATPase is sensitive and labile for this interaction could be disrupted by the addition of ATP. This may also be a possible reason for why we couldn't detect the enzyme activity in vitro.

mEAK7 had been shown to serve as an activator of mTOR signaling since it facilitates the docking of mTOR on the lysosome

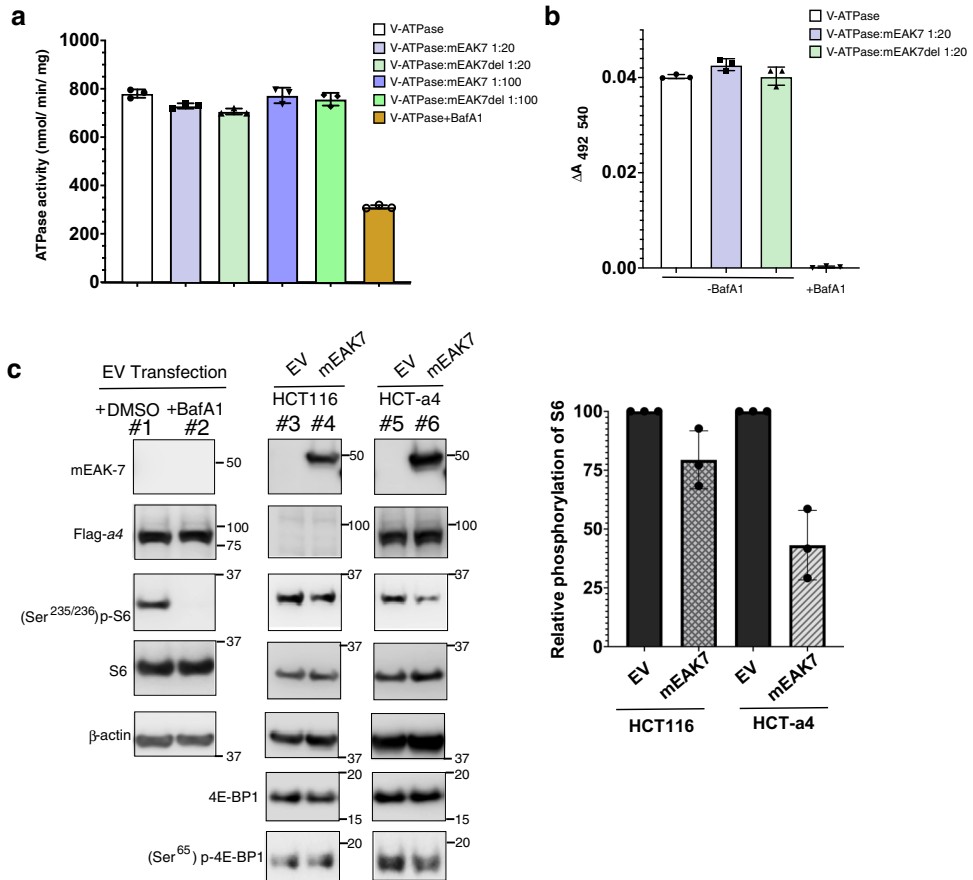

**Fig. 4 mEAK7 doesn't affect the enzyme activity in vitro and inhibits mTOR signaling in HCT-a4 cells. a** NADH-coupled ATPase assay. The concentration of bafilomycin A1 in the assay was 1 μM. Data are mean ± s.d. (n = 3 per group). This experiment was repeated three times independently. Source data are provided as a Source Data file. **b** Proton translocation assay. V-ATPase was incubated with 20 molar excess mEAK7 / mEAK7del for 2 h before assay. The final concentration of bafilomycin A1 in the assay was 10 nM. Data are mean ± s.d. (n = 3 per group). This experiment was repeated three times independently. Source data are provided as a Source Data file. **c** The phosphorylation of S6 is repressed by overexpression of mEAK7 in HCT-a4 cells (left). HCT-a4 cells were transfected with empty vector (EV) or wild-type mEAK7, cultured in low glucose DMEM, 5%FCS for 36–48 h and collected for immunoblot analysis. Bafilomycin A1 (solubilized in DMSO) was supplemented into the EV transfected cells at the final concentration of 2 μM for 2 h before collection. β-actin was used as a loading control. Molecular standards are indicated on the right side of the blots. Quantification of the bands of phosphorylation of S6 by normalization against ribosomal protein S6 (right). The black columns represent the amount of phosphorylation of S6 transfected with EV, which is designated as 100%. Gray columns represent the relative amount of phosphorylation of S6 transfected with mEAK7 in HCT116 cell and HCT-a4 cell, respectively. Error bars represent mean ± s.d. (n = 3), n indicates three independent experiments. Samples derive from the same experiment and the blots were processed in parallel. Source data are provided as a Source Data file. Uncropped blot images are provided in Supplementary Fig. 12.

surface and triggers the phosphorylation of S6 kinase 2 and 4E-BP1[27]. An immunoprecipitation assay demonstrated that the TLD and CTD, which binds to V-ATPase, are also essential for mEAK7 to bind mTOR and to further stimulate mTOR signaling[27]. In contrast, we found that mEAK-7 attenuated mTOR signaling when mEAK-7 is overexpressed in HCT116 cells that stably express subunit a4. The discrepancy in this result may be accounted for by the difference between cell lines. Particularly, the expression level of V-ATPase or other regulators may affect the result. The over-expressed *a4* cell line we used may represent some cells with abnormally high activity of V-ATPase, therefore the inhibition of V-ATPase is favorable for preventing excessive signaling. The different results suggest that mEAK7 may be a multifunctional protein. It is possible that mEAK7 may function as a mTOR signaling modulator–initializing this pathway via engaging with mTOR while concomitantly preventing excessive signaling via V-ATPase inhibition. Alternatively, mEAK7-mediated mTOR regulation may depend on the cell type. Further investigations on mEAK7 are poised to dissect its physiological role and reveal the regulatory relationship between mTOR signaling and V-ATPase activity.

## Methods

### Generation of Expi-a4 and HCT-a4 cell line.
The ATP6V0a4 stable cell line was obtained using a similar method as a previous protocol[23]. Briefly, Expi293F cells (Thermo Fisher Cat #A14527) were plated in 10 cm dishes and maintained in high glucose DMEM (Gibco) with 10% FBS and 1% PS at 37 °C with 5% CO₂. These cells were then transfected with a pcDNA3.1 vector encoding human ATP6V0a4 (subunit *a4*) with C-terminal 2x FLAG tag using transient 2020 (Mirus). After 24 h post transfection, the cells were selected by Geneticin at 800 μg/ml for 2–3 weeks with media changed every 3 days. After clear clones showed up, 30 colonies were enlarged, and detected for the *a4* expression by anti-Flag using western blotting. Five colonies of Expi-a4 cell line that showed higher expression level of *a4* protein by western blotting were combined. Finally, culture medium of Expi-a4 was changed to Expi293 Expression Medium (Thermo Fisher Cat #A1435101) for larger suspension culture.

For the HCT-a4 stable cell line, HCT116 cells (a human colon cancer cell line) were plated in 10 cm dishes and maintained in low glucose DMEM (Gibco) with 5% FBS and 1% PS at 37 °C with 8.8% CO₂. These cells are transfected with the same plasmid above and were selected by Geneticin at 600 μg/ml for 2–3 weeks with media changed every 3 days. Then one colony of HCT-a4 cell line with highest *a4* expression level was selected. By isolating a monoclonal cell population using limiting dilution method, the HCT-a4 monoclonal stable cell line was obtained after 6 weeks.

### Purification of V-ATPase from Expi-a4 cell line.
The suspension Expi-a4 cell line was maintained in Expi293 Expression Medium at 37 °C with 8% CO₂.

pCAG-ALDOA (UniProtKB: P04075) were first transfected into the suspension Expi-a4 cell line using PEI at the density of $1.5 \times 10^6$ cells/ml. After 2 days, about 1.6 L cells were harvested by centrifugation at 1,952 g for 15 min. The cell pellets were suspended with equal volume lysis buffer containing 20 mM Tris-HCl, pH 7.5, 150 mM NaCl, 10 μM D-Fructose 1,6-bisphosphate (FBP), 1 mM CaCl₂, 5 μg/mL leupeptin and 1 mM phenylmethanesulfonylfluoride (PMSF). The cells were then lysed by a Dounce homogenizer and centrifuged at 700 g for 5 min to remove the cell debris, then the pellet was resuspended by a Dounce homogenizer and centrifuged again. The supernatant from these two steps were combined and centrifuged in a T647.5 rotor at 14,629 g for 1 h at 4 °C, then the pellet was resuspended in 10 ml of 10 mg/ml membrane scaffold protein MSP1E3D1[35] in lysis buffer with PMSF, leupeptin and additional 1% N-Dodecyl-β-D-maltoside (DDM) and rotated for 45 min. The lysate was centrifuged at 16,000 g for 15 min, and the supernatant was incubated with 0.4 g/ml Bio-Beads for 2 h. The mixture was poured into a column, and the flow through was collected and mixed with 1.5 ml Flag resin for 1.5 h. The Flag resin was washed with 10 CV wash buffer (20 mM Tris-HCl, pH 7.5, 150 mM NaCl and 10 μM FBP, 1 mM CaCl₂) and eluted with wash buffer containing 200 μg/ml 3X FLAG peptide. The sample was concentrated to 600 μl and was loaded on to a glycerol gradient (11 ml, 20%-50%) prepared in wash buffer. The gradient was centrifuged at 176,433 g for 14 h and fractions were collected. Each fraction containing 820 μl was collected and analyzed by Coomassie blue staining. The fractions containing V-ATPase were concentrated and further purified by gel filtration using a Superose 6 10/300 column (GE Healthcare) pre-equilibrated with buffer containing 20 mM Tris-HCl, pH 7.5, 150 mM NaCl. The peak fractions were collected and concentrated to 1.8 mg/ml for cryo-EM grid preparation.

**EM sample preparation and imaging for Cryo-TEM**. For the human V-ATPase sample assembly in nanodiscs, freshly purified V-ATPase was applied to Quantifoil R1.2/1.3 400 mesh Au holey carbon grids (Quantifoil), then was blotted and plunged into liquid ethane for flash freezing using a Vitrobot Mark IV (FEI). The grids were imaged in a 300 kV Titan Krios (FEI) with a Gatan K3 Summit direct electron detector (Gatan) in super-resolution mode using the data collection software Serial EM. Dark-subtracted images collected at super-resolution mode were first normalized by gain reference and binned twofold, which resulted in a pixel size of 0.842 Å with a counted rate of 8.5 electrons per physical pixel per second. Images were recorded for 5 s exposures in 50 subframes with a total dose of ~60 electrons per Å².

To obtain the complex of *hs*mEAK7-*bt*V-ATPase, the bovine V-ATPase was purified using an established protocol[14,16]. V-ATPase at 4 mg/ml was mixed with a ~10 ×molar excess of human mEAK7 for 1 h prior to grid preparation. The sample was applied to Quantifoil R1.2/1.3 400 mesh Au holey carbon grids (Quantifoil), and then blotted and plunged into liquid ethane for flash freezing using a Vitrobot Mark IV (FEI). The grids were imaged in a 300 kV Titan Krios (FEI) with a Gatan K3 Summit direct electron detector (Gatan) in super-resolution mode using the data collection software Serial EM. Dark-subtracted images collected at super-resolution mode were first normalized by gain reference and binned twofold, which resulted in a pixel size of 0.83 Å with a counted rate of 8.5 electrons per physical pixel per second. Images were recorded for 5 s exposures in 50 subframes with a total dose of ~60 electrons per Å².

**Imaging processing and 3D reconstruction for Cryo-TEM**. For the human V-ATPase sample assembly in nanodiscs, the images were collected by two sessions. In session 1, 3978 images were collected. Dark-subtracted images were first normalized by gain reference and then binned twofold which resulted in a pixel size of 0.842 Å. Motion correction and gain reference was performed using the program MotionCor2[36]. The contrast transfer function (CTF) was estimated using CTFFIND4[37]. Low-quality images and false-positive particles were removed manually. The V-ATPase templates for automatic picking was from the apo-V-ATPase data[14], and after auto-picking by RELION-3[38], 696,668 particles of V-ATPase were extracted, then the particles were classified by 2D classification by cryoSPARC[39]. We used our previously determined cryo-EM structure of apo-V-ATPase low-pass filtered to 40 Å as the initial model for hetero refinement by cryoSPARC. An extremely weak extra density near $V_1$ domain of V-ATPase was observed at the low threshold of some classes. One of the classes with stronger density was selected and subjected to the second hetero refinement by cryoSPARC. In this classification, state 2 and state 3 of V-ATPase could be distinguished. For state 2, another 2D classification and hetero-refinement were performed to pick the class with the stronger external part. As the density of this external part was not effectively enhanced through multiple classifications, the class with 97,490 particles, which contains an external part in the $V_1$ domain was selected and subjected to RELION-3 for further 3D classification. To better separate the external part bound V-ATPase and apo-V-ATPase, we masked the external part and performed a no image alignment Class3D, about 17.9% population with external part was selected and refined by 3D refinement. Session 2 was processed using a similar strategy. 4671 images were collected during session 2. After auto-picking by RELION-3, 697,674 particles were extracted, then the particles were classified by 2D classification and hetero refinement by cryoSPARC. The class which showed state 2 of V-ATPase was selected and subjected to RELION-3 for 3D classification and a second 3D classification with a local mask, which was generated from the map in session 1. After 3D refinement, Bayesian polishing was performed for the particles from two sessions. The final refinement, which used 24,984 particles after Bayesian polishing, provided a 4.08 Å map and resolution was estimated using the Fourier shell correlation (FSC) 0.143 criterion by RELION-3. For *hs*mEAK7-*bt*V-ATPase complex, 6772 images were collected. Dark-subtracted images were first normalized by gain reference and then binned twofold which resulted in a pixel size of 0.83 Å. The previous data processing steps are similar to the human V-ATPase sample, and after auto-picking by RELION-3[38], 604,862 particles were extracted, then the particles were classified by 2D classification by cryoSPARC[39]. We used our previously determined cryo-EM structure of apo-V-ATPase state 1, state 2 and *hs*V-ATPase-*hs*mEAK7 complex as the initial models for hetero refinement by cryoSPARC. The good class with 34,841 particles which showed the obvious feature of mEAK7 was selected for further 3D classification. This classification separated state 1 and state 2 with mEAK7. Then classes with mEAK7 feature were selected and subjected to RELION-3 for another 3D classification, two best classes with about 13,841 particles were selected for 3D refinement, yielding a 4.15 Å map after postprocessing. Then Bayesian polishing was performed for the particles and finally provided a 4.11 Å map and resolution was estimated using the Fourier shell correlation (FSC) 0.143 criterion by RELION-3.

Owing to the flexibility of $V_1$ and $V_o$, $V_o$ was not well resolved. We referred to our previous method[16], and several focused refinements with different masks were attempted to get a high-quality local map. For the human V-ATPase assembly in nanodiscs, these focused refinements including subunits $A_3B_3DE_3FG_3$ and mEAK7, c-ring, and subunits $CFHV_o$ provided good local maps, especially for $V_o$ (Supplementary Figs. 2,3 and 4). After postprocessing, these refinements gave a resolution of 3.78 Å, 3.78 Å, and 4.08 Å, respectively. These focused maps can be used to generate a composite map for refinement. For *hs*mEAK7-*bt*V-ATPase complex, the focused refinement was performed only on the $V_1$ domain including subunits $A_3B_3DE_3FG_3$ and mEAK7, and this refinement yielded a 3.73 Å map after postprocessing.

**Model construction, model refinement and validation**. To generate a composite map for refinement of the mEAK7-bound *hs*V-ATPase, several focused maps that have been described above were combined and aligned using phenix.combine_focused_maps which would coalesce the best parts of several maps together as our previous paper reported. Most of the subunit models were based on the state 2 structure of V-ATPase (PDB: 6WM3) (https://doi.org/10.2210/pdb6WM3/pdb) from HEK293T cells, including subunit A, B2, C1, D, E1, F, G1, H, *b*, *c*, *d1*, *e1*, RNAseK, Ac45 and PRR. The models of subunit *a4* and mEAK7 were predicted by AlphaFold[26]. The overall structures of state 2, *a4* and mEAK7 were docked into the cryo-EM map of mEAK7-bound V-ATPase separately using Chimera[40] to generate the initial model, and then manually adjusted using COOT[41]. For *hs*mEAK7-*bt*V-ATPase complex, the models of human subunits A, B, D, E, G, and mEAK7 from mEAK7-bound *hs*V-ATPase were docked into the map from the focused refinement of $V_1$ domain using Chimera to generate the initial model, the residues were mutated into bovine V-ATPase manually, then the model was adjusted using COOT. All the models were refined in real space using PHENIX[42] with secondary-structure restraints and stereochemical restraints[43]. For cross-validations, the final model was refined against one of the half maps generated by 3D auto-refine and the model vs. map FSC curves were generated in the comprehensive validation module in PHENIX. MolProbity[44] and PHENIX were used to validate the final models. Local resolutions were estimated using RELION-3. Structure figures were generated using PyMOL (http://www.pymol.org), Chimera[40] and ChimeraX[45].

**Mass spectrometry identification**. The V-ATPase samples were cut from SDS-PAGE and digested overnight with trypsin (Pierce) following reduction and alkylation with DTT and iodoacetamide (Sigma–Aldrich). The samples then underwent solid-phase extraction cleanup with an Oasis HLB plate (Waters) and were subsequently dried and reconstituted into 10 ul of 2% ACN, 0.1% TFA. 2 ul of these samples were injected onto a QExactive HF mass spectrometer coupled to an Ultimate 3000 RSLC-Nano liquid chromatography system. Samples were injected onto a 75 um i.d., 15-cm long EasySpray column (Thermo) and eluted with a gradient from 0-28% buffer B over 90 min with a flow rate of 250 nl/min. Buffer A contained 2% (v/v) ACN and 0.1% formic acid in water, and buffer B contained 80% (v/v) ACN, 10% (v/v) trifluoroethanol, and 0.1% formic acid in water. The mass spectrometer operated in positive ion mode with a source voltage of 2.2 kV and an ion transfer tube temperature of 275 °C. MS scans were acquired at 120,000 resolution in the Orbitrap and up to 20 MS/MS spectra were obtained for each full spectrum acquired using higher-energy collisional dissociation (HCD) for ions with charges 2–8. Dynamic exclusion was set for 20 s after an ion was selected for fragmentation.

Raw MS data files were analyzed using Proteome Discoverer v2.4 (Thermo), with peptide identification performed using Sequest HT searching against the human mEAK7 protein database from UniProt. Fragment and precursor tolerances of 10 ppm and 0.02 Da were specified, and three missed cleavages were allowed. Carbamidomethylation of cysteine was set as a fixed modification, with oxidation of Met set as a variable modification. The false-discovery rate (FDR) cutoff was 1% for all peptides.

**Purification of human mEAK7 protein**. The cDNA of human mEAK7 (Nucleotide Accession: BC060844.1) was cloned into the pET-28a vector (Novagen) with a N-terminal 6 x His tag and expressed in *E.coli* BL21 (DE3) cells. The transformed cells were grown at 37 °C in LB medium and then induced at $OD_{600} = 0.8$ with 0.5 mM IPTG for 3.5 h at 22 °C. The cells were harvested and disrupted by sonication in buffer C, containing 20 mM HEPES pH 7.5, 150 mM NaCl, 1 mM PMSF and 5 µg/ml leupeptin. After the cell lysates were centrifuged at 38,759 g for 40 min, the supernatant was purified by affinity chromatography using a Ni-NTA column (QIAGEN) with buffer C supplemented with 20 mM imidazole and 300 mM imidazole serving as washing and elution buffer. Then mEAK7 was further purified by gel filtration using an Surperdex 200 increased 10/300 column pre-equilibrated with buffer D (10 mM Tricine pH 7.0, 150 mM KCl, 3 mM $MgCl_2$). The elution peak was concentrated and stored for future use. The mEAK7del (1-413) construct was expressed and purified in the same way with full-length mEAK7 protein.

**Bovine V-ATPase purification and reconstitution into proteoliposomes**. The bovine V-ATPase was purified from bovine brain using an established protocol[22]. Briefly, the membrane vesicles prepared from bovine brain were suspended with 0.75 M Tris-HCl pH 8.0 to strip clathrin. After several separation steps, the pellets were solubilized with 1% (v/v) $C_{12}E_9$, 10 mM Tris-MES pH 7.0. After centrifugation, the supernatant was purified with a hydroxyapatite column. The good fractions were loaded into a glycerol gradient (12 ml, 15–30%) for further purification after saturated ammonium sulfate precipitation. The good fractions from the glycerol gradient were used for preparing proteoliposomes and the ATPase assay.

The bovine V-ATPase was reconstituted into proteoliposomes containing phosphatidylcholine (PC), phosphatidylethanolamine (PE), Phosphatidylserine (PS), and cholesterol at a weight ratio of 40:26.5:7.5:26[16,46] by the cholate dilution, freeze-thaw method. For each reaction, 1 µg V-ATPase was mixed with 300 µg liposomes (liposome stock at 50 mg/ml) and reconstitution buffer to yield the protein-lipid mixture with glycerol, sodium cholate, KCl, $MgCl_2$ and DTT at final concentrations of 10% (v/v), 1%, 150 mM, 2.5 mM, and 2 mM, respectively. The reconstitution mixtures were incubated at room temperature for 1 h, followed by freezing in liquid $N_2$ for 5 min. Then the reconstitution mixture was thawed on ice for the proton translocation assay.

**Measurement of ATP-driven proton translocation assay**. V-ATPase purified from bovine brain was used for the ATP-driven proton translocation assay. V-ATPase was incubated with ~20 molar excess buffer / mEAK7 / mEAK7del for 2 h on ice before each assay. For each reaction, 20 µl proteoliposome mixture was added to 1.6 ml of the proton-pumping assay buffer containing 10 mM Tricine pH 7.0, 6.7 µM acridine orange, 3 mM $MgCl_2$, and 150 mM KCl[16]. The reaction was initiated by the addition of 1.3 mM ATP pH 7.0, 1 µg/ml valinomycin. The assay was conducted in a SLM-Aminco DW2C dual wavelength spectrophotometer and the activity was registered as $\Delta A_{492-540}$[17]. The concentration of bafilomycin A1 in the assay was 10 nM. All experiments were performed three times on different days, and the results were visualized using GraphPad Prism8.

**ATPase activity assay**. The ATPase activity was measured using an NADH consumption coupled method[47]. Absorbance at 340 nm was monitored to measure the concentration of NADH that was coupled to that of ATP. The assay was performed at 37 °C in a 96-well plate and the final reaction contained 5 nM V-ATPase, 50 mM Tris-MES pH 7.0, 30 mM KCl, 3 mM $MgCl_2$, 1 mM DTT, 4 mM phosphoenolpyruvate, 60 µg/mL pyruvate kinase, 33 µg/mL lactate dehydrogenase, 0.2 mM NADH, and 2 mM ATP with a total reaction volume of 100 µL. The absorbance was measured every 20 s for 40 min using a BioTek Synergy Neo plate reader. The $V_{max}$ of absorbance change (min$^{-1}$) was calculated by the built-in software using 20–30 points in the linear region, which was converted to the rate of ATP hydrolysis (nmol ATP·min$^{-1}$·mg protein$^{-1}$) by dividing the product of the extinction coefficient of NADH, the length of the light path, and the concentration of V-ATPase.

**Immunoblot analysis of S6 phosphorylation**. HCT-a4 cells were seeded on the 60-mm dish and transfected with 3 µg empty vector pCAG (EV) or pCAG-mEAK7 using FuGENE 6 (Promega) at 40–60% confluency. For the DMSO and bafilomycin A1 treatment, bafilomycin A1 (solubilized in DMSO) was added to a final concentration of 2 µM 2 h before harvest, while DMSO of the same volume was added as a control. Cells were collected with a cell scraper after 36–48 h post transfection and lysed with a cold NP40 lysis buffer (50 mM Tris-HCl, pH 8.0, 150 mM NaCl, 1% NP-40 and EDTA-free protease inhibitor (Roche)) for 15 mins. The cell lysates were centrifuged at 18,407 g for 5 min, then the supernatants were used for detection by western blotting. About 18 µg sample were applied to the Bolt 4–12% gradient gels. After electrophoresis, the proteins were transferred to nitrocellulose filters, which were then incubated with the indicated primary antibody. All primary antibodies were listed as follows: Anti-KIAA1609 polyclonal antibody (Abnova, PAB23736, 1:500 dilution), β-actin Mouse mAb (CST, 3700 S, 1:1000 dilution), S6 ribosomal protein Mouse mAb (CST, 2317 S, 1:1000 dilution), Phospho-S6 ribosomal protein (Ser235/236) Rabbit mAb (CST, 4858 S, 1:2000 dilution), 4E-BP1 Antibody (CST, 9452, 1:1000 dilution), Phospho-4E-BP1(Ser65) Antibody (CST, 9451 S, 1:1000 dilution), Anti-DDDDK-tag mAb (MBL, M185-3L,

1:3000 dilution). The bound antibodies were then visualized by chemiluminescence (SuperSignal West Pico Chemiluminescent Substrate, Thermo Scientific, Waltham, MA) after a 1 h incubation with either horse anti-mouse IgG (1:3000 dilution) or goat anti-rabbit IgG (1:3000 dilution) conjugated to HRP. The images were scanned using an Odyssey FC Imager (Dual-Mode Imaging System) and analyzed using Image Studio ver. 5.0 (LI-COR Biosciences, Lincoln, NE). All experiments in Fig. 4c were repeated three times on different days. Similar results were obtained. Images were analyzed, and band intensities were quantified using ImageJ2. Bar graphs were generated by Prism8 (GraphPad). The uncropped blotting images are shown in Supplementary 12.

**Fluorescence microscopy**. 450,000 HCT-a4 cells were plated on the 6 well plate with glass coverslips and transfected with 1.5 µg LAMP1-mCherry using TransIT-2020 (Mirus Bio) at 20%–30% confluency. After 36 h, the slides were rinsed with PBS and fixed with 4% paraformaldehyde at room temperature for 15 min. Then the slides were permeabilized with 0.05% Triton X-100 in PBS for 2 min and was incubated sequentially with Mouse anti-FLAG antibody (MBL, M185-3L, 1:10,000 dilution) and goat anti mouse IgG Alexa Fluor 488 (Thermo Fisher Scientific, A-11001, 1:2000 dilution). The cells were next washed with phosphate-buffered saline (PBS) and stained with 4′,6-diamidino-2-phenylindole (DAPI; 0.1 µg/ml) in the dark for 5 min at room temperature. After this, the coverslips were mounted on glass slides in Shandon Immu-mount (Thermo Fisher Scientific). The fluorescence images were acquired using a Plan-Apochromat ×63/1.4 oil differential interference contrast objective (Zeiss, Oberkochen, Germany), a Zeiss LSM 800 microscope (Zeiss), and ZEN Blue 2.5 software (Zeiss).

**Reporting summary**. Further information on research design is available in the Nature Research Reporting Summary linked to this article.

## Data availability

The 3D cryo-EM density maps of mEAK7-bound *hs*V-ATPase and $V_1$ region of *bt*V-ATPase complex in complex with *hs*mEAK7 have been deposited in the Electron Microscopy Data Bank under the accession numbers EMD-26623 (*hs*mEAK7-*hs*V-ATPase structure) and EMD-26622 (*hs*mEAK7-*bt*V-ATPase structure). Atomic coordinates for the atomic model of mEAK7-bound *hs*V-ATPase and *hs*mEAK7-*bt*$V_1$-region have been deposited in the Protein Data Bank under the accession number 7UNF (*hs*mEAK7-*hs*V-ATPase structure) and 7UNE (*hs*mEAK7-*bt*V-ATPase structure). The mEAK7 gene (Nucleotide Accession: BC060844.1) was purchased from the Center of Human Genetics in UT Southwestern Medical Center. Additional data supporting the findings in this study are provided as source data and supplementary information to this manuscript.

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

## Acknowledgements

The data were collected at the UT Southwestern Medical Center Cryo-EM Facility (funded in part by the CPRIT Core Facility Support Award RP170644). We thank Y. Sun for assistance in data collection, P. Schmiege and E. Debler for editing the manuscript, A. Lemoff at the UT Southwestern Proteomics Core for mass spectrometry analysis, Y. Yu and J. Tian for sharing the antibodies with us. This work was supported by NIH grant R01 HL072304, P01 HL020948, R01 GM135343, and Welch Foundation (I-1957) (to X.L.). X.L. is a Damon Runyon-Rachleff Innovator supported by the Damon Runyon Cancer Research Foundation (DRR-53S-19) and a Rita C. and William P. Clements Jr. Scholar in Biomedical Research at UT Southwestern Medical Center.

## Author contributions

R.W. and X.L. conceived the project and designed the research with X.-S.X., R.W. and Y.Q. performed the study. All authors analyzed the data and contributed to manuscript preparation. X.L. wrote the manuscript.

## Competing interests

The authors declare no competing interests.
