## [Peer Review File · Nature Communications]

Molecular Basis of mEAK7-Mediated Human V-ATPase RegulationREVIEWER COMMENTS

Reviewer #1 (Remarks to the Author):

This is another beautiful work from Xiaochun Li's group that describes an important structural model for the vATPase-mEAK7 complex. Combining cryo-EM, biochemical, and cell biology approaches, the authors convincingly identified the protein corresponding to a previously unknown density during v-ATPase purification. They further built an atomic model, mapped the interaction between mEAK7 and v-ATPase, and showed that mEAK7 inhibits v-ATPase activity in a specific cell line. These results are novel and are of great importance to the field of lysosome research. I only have one minor suggestion and I fully support publication on Nature Communications.

Minor concern:

The authors should include a paragraph at the end of the Introduction section to briefly summarize their results. This will help guide the readers.

Reviewer #2

Please see attached.

In this manuscript, Wang et. al. found that a lysosomal protein called mEAK-7 binds and regulates mammalian V-ATPases through cryo-EM structure determination. mEAK-7 was found to be involved in dauer formation in nematodes and mTOR signaling in mammals. Unexpectedly, the high-resolution cryo-EM structure in this manuscript revealed that mEAK-7 interacts with subunits A, B, E, and D and thereby locks V-ATPases in state 2. Thus, mEAK-7 was hypothesized to inhibit the activity of V-ATPases. However, the *in vitro* activity assay showed that mEAK-7 failed to block the ATP hydrolysis activity and the proton transfer activities of V-ATPases. Then the authors further utilized cell-based assays to show that mEAK-7 attenuated mTOR signaling in HCT116 cells. Overall, this paper represents a new step in the field and the data is very solid to support the conclusions. Some minor issues listed below should be addressed before the manuscript being published.

1. When measuring the ATPase activity and proton transfer of V-ATPases, the authors used human mEAK-7 and bovine V-ATPases. However, the author did not discuss the sequence conservation of between human and bovine mEAK-7.

- I would suggest to adding a sequence alignment of mEAK-7 in the supplemental figures to justify that human mEAK-7 is equivalent to the bovine one.
2. mEAK-7 was shown to be a positive regulator of mTOR signaling by others. In contrast, here the author showed that mEAK-7 attenuated mTOR signaling. This discrepancy was not sufficiently addressed. The authors should revise the discussion part to better address this issue.
 3. In the competing bioRxiv manuscript (Tan, et. al., 2021), the authors showed that in cells overexpression of mEAK-7 does not lead to the pH changes of lysosomes. This is consistent with the findings in this manuscript by Wang et. al. I think that their work should be cited and discussed in the section of “mEAK-7 may act as a V-ATPase inhibitor” of Page 9.

Reviewer #3 (Remarks to the Author):

This manuscript describes a novel cryo-EM structure of a TLDC family protein, mEAK-7, bound to a human V-ATPase. Recent evidence indicates that some TLDC proteins bind to V-ATPases and may serve as modulators. The authors use a recent strategy of stably overexpressing a FLAG-tagged α -subunit isoform, $\alpha 4$, in HEK293 cells to isolate assembled V-ATPases containing the $\alpha 4$ subunit along with endogenous subunits and modulators from the cell line. A subset (5%) of the V-ATPase complexes contains mEAK-7 bound to several V1 subunits in a position that should inhibit ATP hydrolysis at the state 2 position in the rotary cycle. The authors seek further support for mEAK-7 as a V-ATPase inhibitor by expressing mEAK-7 in bacteria and adding it to the highly conserved bovine V-ATPase, but observe no inhibition of ATP hydrolysis or proton pumping. Inhibition of V-ATPase activity with established inhibitors such as bafilomycin is known to inhibit mTOR activity, and previous data suggests that mEAK-7 activates mTOR activity by stimulating its recruitment to the lysosomal membrane. The authors overexpress mEAK-7 in HCT111 cells, with and without stable overexpression of the V-ATPase $\alpha 4$ isoform. Curiously, they see partial inhibition of mTOR activity with mEAK-7 overexpression, but only in the $\alpha 4$ -containing cell line.

The structure is interesting and well-supported by the data. It is particularly interesting that mEAK-7 appears to trap the V-ATPase in a different rotational position than another TLDC protein, Oxr1. This supports the idea that this family has a general role in modulating V-ATPase activities by binding to V1, but can exert these roles in structurally distinct ways. However, additional experiments to probe the role of mEAK-7 are too preliminary to allow definitive conclusions.

1. There is very little evidence that TLDC is a V-ATPase inhibitor beyond the structure. The failure of expressed mEAK-7 to inhibit the bovine enzyme would certainly argue against mEAK-7 as an inhibitor. However, no evidence is provided that mEAK-7 actually binds to the bovine enzyme. One explanation that is not considered is that the N-terminal myristoylation of mEAK-7, which would be missing from

bacterially expressed protein, might help to position it at the membrane and assist in its binding to V1. The N-terminal domain of mEAK-7 is poorly resolved in the structure, but it is likely that it was membrane-bound in the original HEK293 cells used for isolation.

2. The proposal that mEAK-7 functions “between” the V-ATPase and mTOR to affect mTOR signaling is intriguing but is not strongly supported by the data. The evidence that overexpression of mEAK-7 inhibits phosphorylation of downstream targets, Figure 4C, should be supported with further quantitation and statistical analysis since the mEAK-7 effect is not that strong. Furthermore, the inhibition only seems to occur in the presence of overexpressed a4, even though V-ATPases with other a subunit isoforms would be present in the HCT111 strain. (Since there is no evidence of binding to the a subunit, the a subunit isoform shouldn’t matter.) It would help to model to demonstrate that the overexpressed a4 subunit is at least in the lysosome, where it could interact with mEAK-7 and affect mTOR signaling. This could be done by immunofluorescence.

In summary, this is a promising but incomplete story as it stands. The strategy for isolation of novel V-ATPase modulators is innovative. The structure is very nice and suggests intriguing mechanistic possibilities, but neither a role for mEAK-7 as a true V-ATPase inhibitor nor a role connecting V-ATPase and mTOR are established by the current data.

Reviewer #4 (Remarks to the Author):

The authors present cryo-EM data showing mEAK7 interactions with the V1 domain of assembled V-ATPase. The structural study is sound but unfortunately, the study does not provide answers for what is the functional significance of this interaction. The experimental data does not support the conclusion that mEAK7 is an inhibitor of V-ATPase as the authors suggest.

Ref # 32 shows that the findings of this study are published in BioRxiv (2021) “CryoEM of endogenous mammalian V-ATPase interacting with the TLDC protein mEAK-7”

(doi: <https://doi.org/10.1101/2021.11.03.466369>). In ref 32, the authors also concluded that exogenous mEAK-7 does not inhibit V-ATPase.

Other revisions:

The authors need to cite recent publications relevant to this study which have shown interaction of TLD containing proteins with V1 and V1Vo:

1) Sci Rep. 2021 Nov 22;11(1):22654. doi: 10.1038/s41598-021-01809-y.

2) EMBO J. 2022 Feb 1;41(3):e109360. doi: 10.15252/embj.2021109360. Epub 2021 Dec 17.

Line 123, the statement that “The TLD facilitates mEAK-7 docking to the V-ATPase and then allows the CTD to insert into the cavity in the middle of the ABsemi heterodimer “ seems speculative at this time.

Response to Reviewers:

Reviewer #1

This is another beautiful work from Xiaochun Li's group that describes an important structural model for the vATPase-mEAK7 complex. Combining cryo-EM, biochemical, and cell biology approaches, the authors convincingly identified the protein corresponding to a previously unknown density during v-ATPase purification. They further built an atomic model, mapped the interaction between mEAK7 and v-ATPase, and showed that mEAK7 inhibits v-ATPase activity in a specific cell line. These results are novel and are of great importance to the field of lysosome research. I only have one minor suggestion and I fully support publication on Nature Communications.

Minor concern:

The authors should include a paragraph at the end of the Introduction section to briefly summarize their results. This will help guide the readers.

Response: Point accepted. We have added the summarized paragraph in the Introduction, please see lines 53-60.

The authors thank this reviewer for her/his time and constructive comments.

Reviewer #2

In this manuscript, Wang et. al. found that a lysosomal protein called mEAK-7 binds and regulates mammalian V-ATPases through cryo-EM structure determination. mEAK-7 was found to be involved in dauer formation in nematodes and mTOR signaling in mammals.

Unexpectedly, the high-resolution cryo-EM structure in this manuscript revealed that mEAK-7 interacts with subunits A, B, E, and D and thereby locks V-ATPases in state 2. Thus, mEAK-7 was hypothesized to inhibit the activity of V-ATPases. However, the in vitro activity assay showed that mEAK-7 failed to block the ATP hydrolysis activity and the proton transfer activities of V-ATPases. Then the authors further utilized cell-based assays to show that mEAK-7 attenuated mTOR signaling in HCT116 cells. Overall, this paper represents a new step in the field and the data is very solid to support the conclusions. Some minor issues listed below should be addressed before the manuscript being published.

1. When measuring the ATPase activity and proton transfer of V-ATPases, the authors used human mEAK-7 and bovine V-ATPases. However, the author did not discuss the sequence conservation of between human and bovine mEAK-7. I would suggest to adding a sequence alignment of mEAK-7 in the supplemental figures to justify that human mEAK-7 is equivalent to the bovine one.

Response: Point accepted. The sequence identity of the interaction subunits A, B, D, E of V-ATPase is over 97% between bovine and human V-ATPase, and the residues participating in mEAK7 binding are exactly the same. In contrast, human mEAK7 and bovine mEAK7 only have a ~67% sequence identity, and the TLD and CTD domains are not well conserved (Supplementary Fig. 10d). To verify whether human mEAK7 can bind bovine V-ATPase as well as human V-ATPase, we determined the cryo-EM structure of *hsmEAK7-btV-ATPase* complex, and the structural analysis showed human mEAK7 could form a complex with bovine V-ATPase similar to human V-ATPase (Supplementary Figs. 9 and 10).

2. mEAK-7 was shown to be a positive regulator of mTOR signaling by others. In contrast, here the author showed that mEAK-7 attenuated mTOR signaling. This discrepancy was not sufficiently addressed. The authors should revise the discussion part to better address this issue.

Response: Point accepted. The experiment results seem to be different between us. They (Science Advances 4, eaao5838) found an increased phosphorylation of S6 when they overexpressed mEAK-7 with C-terminal HA tag in four different cell lines, while we showed that mEAK-7 attenuated mTOR signaling when non-tagged mEAK-7 is overexpressed in HCT116 cells that stably expresses subunit *a4*. From the structural analysis, we observed the C-terminus inserted into V-ATPase, which is indispensable in blocking the rotary of V-ATPase, the extra C-terminal HA tag probably hinder the binding. We speculate that the overexpressed *a4* cell line may represent some cells with an abnormally high activity of V-ATPase, the inhibition of V-ATPase is favorable for preventing excessive signaling. We have added this discussion in the main text, please see lines 246-253.

3. In the competing bioRxiv manuscript (Tan, et. al., 2021), the authors showed that in cells overexpression of mEAK-7 does not lead to the pH changes of lysosomes. This is consistent with

the findings in this manuscript by Wang et. al. I think that their work should be cited and discussed in the section of “mEAK-7 may act as a V-ATPase inhibitor” of Page 9.

Response: Point accepted. We have added a discussion about the conclusion of this bioRxiv paper in lines 235-240.

The authors thank this reviewer for her/his time and constructive comments.

Reviewer #3

This manuscript describes a novel cryo-EM structure of a TLDC family protein, mEAK-7, bound to a human V-ATPase. Recent evidence indicates that some TLDC proteins bind to V-ATPases and may serve as modulators. The authors use a recent strategy of stably overexpressing a FLAG-tagged α -subunit isoform, $\alpha 4$, in HEK293 cells to isolate assembled V-ATPases containing the $\alpha 4$ subunit along with endogenous subunits and modulators from the cell line. A subset (5%) of the V-ATPase complexes contains mEAK-7 bound to several V1 subunits in a position that should inhibit ATP hydrolysis at the state 2 position in the rotary cycle. The authors seek further support for mEAK-7 as a V-ATPase inhibitor by expressing mEAK-7 in bacteria and adding it to the highly conserved bovine V-ATPase, but observe no inhibition of ATP hydrolysis or proton pumping. Inhibition of V-ATPase activity with established inhibitors such as bafilomycin is known to inhibit mTOR activity, and previous data suggests that mEAK-7 activates mTOR activity by stimulating its recruitment to the lysosomal membrane. The authors overexpress mEAK-7 in HCT111 cells, with and without stable overexpression of the V-ATPase $\alpha 4$ isoform. Curiously, they see partial inhibition of mTOR activity with mEAK-7 overexpression, but only in the $\alpha 4$ -containing cell line.

The structure is interesting and well-supported by the data. It is particularly interesting that mEAK-7 appears to trap the V-ATPase in a different rotational position than another TLDC protein, Oxr1. This supports the idea that this family has a general role in modulating V-ATPase activities by binding to V1, but can exert these roles in structurally distinct ways. However, additional experiments to probe the role of mEAK-7 are too preliminary to allow definitive conclusions.

1. There is very little evidence that TLDC is a V-ATPase inhibitor beyond the structure. The failure of expressed mEAK-7 to inhibit the bovine enzyme would certainly argue against mEAK-7 as an inhibitor. However, no evidence is provided that mEAK-7 actually binds to the bovine enzyme. One explanation that is not considered is that the N-terminal myristoylation of mEAK-7, which would be missing from bacterially expressed protein, might help to position it at the membrane and assist in its binding to V1. The N-terminal domain of mEAK-7 is poorly resolved in the structure, but it is likely that it was membrane-bound in the original HEK293 cells used for isolation.

Response: Pointe accepted. To verify that human mEAK7 binds to the bovine V-ATPase, we mixed bovine V-ATPase with *E. coli* expressed human mEAK7 at 1:10 molar ratio and collected the cryo-EM data for the complex. The structural analysis showed that human mEAK7 binds to bovine V-ATPase (Supplementary Figs. 9-10). Thus, we can confirm that bovine V-ATPase can be used as a substitute of human V-ATPase for *in vitro* assays.

2. The proposal that mEAK-7 functions "between" the V-ATPase and mTOR to affect mTOR signaling is intriguing but is not strongly supported by the data. The evidence that overexpression of mEAK-7 inhibits phosphorylation of downstream targets, Figure 4C, should be supported with further quantitation and statistical analysis since the mEAK-7 effect is not that strong. Furthermore, the inhibition only seems to occur in the presence of overexpressed $\alpha 4$, even though V-ATPases with other α subunit isoforms would be present in the HCT111 strain. (Since there is no evidence of binding to the α subunit, the α subunit isoform shouldn't matter.) It would

help to model to demonstrate that the overexpressed a4 subunit is at least in the lysosome, where it could interact with mEAK-7 and affect mTOR signaling. This could be done by immunofluorescence.

Response: Point accepted. We have added quantitation and statistical analysis which statistics the phosphorylation level of S6 from 3 independent experiments in Figure 4c. According to referee's suggestion, we have added the immunofluorescence assay in the revision. The immunofluorescence assay shows that the overexpressed a4 subunit could localize in the lysosome (Supplementary Fig. 11).

In summary, this is a promising but incomplete story as it stands. The strategy for isolation of novel V-ATPase modulators is innovative. The structure is very nice and suggests intriguing mechanistic possibilities, but neither a role for mEAK-7 as a true V-ATPase inhibitor nor a role connecting V-ATPase and mTOR are established by the current data.

Response: We hope our responses and additional experimental evidence have addressed the concerns of this reviewer.

The authors thank this reviewer for her/his time and constructive comments.

Reviewer #4

The authors present cryo-EM data showing mEAK7 interactions with the V1 domain of assembled V-ATPase. The structural study is sound but unfortunately, the study does not provide answers for what is the functional significance of this interaction. The experimental data does not support the conclusion that mEAK7 is an inhibitor of V-ATPase as the authors suggest.

Ref # 32 shows that the findings of this study are published in BioRxiv (2021) “CryoEM of endogenous mammalian V-ATPase interacting with the TLDc protein mEAK-7” (doi: <https://doi.org/10.1101/2021.11.03.466369>). In ref 32, the authors also concluded that exogenous mEAK-7 does not inhibit V-ATPase.

Response: We have added a discussion about the conclusion of this bioRxiv paper in lines 235-240. Please also see our response to Reviewer 2’s Point 3.

Other revisions:

The authors need to cite recent publications relevant to this study which have shown interaction of TLD containing proteins with V1 and V1Vo:

- 1) Sci Rep. 2021 Nov 22;11(1):22654. doi: 10.1038/s41598-021-01809-y.
- 2) EMBO J. 2022 Feb 1;41(3):e109360. doi:. Epub 2021 Dec 17.

Response: Point accepted. We have added these citations in the revision (lines 167-170).

Line 123, the statement that “The TLD facilitates mEAK-7 docking to the V-ATPase and then allows the CTD to insert into the cavity in the middle of the ABsemi heterodimer “ seems speculative at this time.

Response: Point accepted. We have changed the sentence as “The specific mechanism of how mEAK7 binds to V-ATPase remains unclear, it is possible that the TLD facilitates mEAK7 docking to the V₁ domain and may allow the CTD to insert into the cavity in the middle of the ABsemi heterodimer” (lines 129-131).

The authors thank this reviewer for her/his time and constructive comments.

REVIEWERS' COMMENTS

Reviewer #2 (Remarks to the Author):

The authors have addressed most of my questions. However, the response to my question regarding the effect of mEAK7 on mTOR signaling is not convincing. Wang et. al. in this manuscript showed that mEAK-7 inhibits mTOR signaling while Nguyen et. al had a paper published in Science advances clearly showed that mEAK-7 promotes mTOR signaling. Wang et. al thought that Nguyen et. al. used a C-terminal HA tagged mEAK-7, which is one of the major reasons leading to the different results. However, the fact is that Nguyen et al. also showed that knockdown of endogenous mEAK-7 in H1299 cells also leads to reduced S6 phosphorylation (see Figure 4I & 4K). Thus, the HA tag cannot explain the discrepancy of the two manuscripts. I hope that the author could revise this part and provide a more reasonable explanation to this discrepancy before the manuscript being accepted for publication.

Reviewer #3 (Remarks to the Author):

The authors have addressed the comments raised in my previous review very well.

One very small change:

On line 238, I believe the word "liable" should actually be "labile".

Response to Reviewers:

Reviewer #2

The authors have addressed most of my questions. However, the response to my question regarding the effect of mEAK7 on mTOR signaling is not convincing. Wang et. al. in this manuscript showed that mEAK-7 inhibits mTOR signaling while Nguyen et. al had a paper published in Science advances clearly showed that mEAK-7 promotes mTOR signaling. Wang et. al thought that Nguyen et. al. used a C-terminal HA tagged mEAK-7, which is one of the major reasons leading to the different results. However, the fact is that Nguyen et al. also showed that knockdown of endogenous mEAK-7 in H1299 cells also leads to reduced S6 phosphorylation (see Figure 4I & 4K). Thus, the HA tag cannot explain the discrepancy of the two manuscripts. I hope that the author could revise this part and provide a more reasonable explanation to this discrepancy before the manuscript being accepted for publication.

Response: Point accepted. We found that mEAK-7 attenuated mTOR signaling when mEAK-7 is overexpressed in HCT116 cells that stably express subunit a4. The discrepancy in this result may be accounted for by the difference between cell lines. Particularly, the expression level of V-ATPase or other regulators may affect the result. We have added this discussion in the revised manuscript.

The authors thank this reviewer for her/his time and constructive comments.

Reviewer #3

The authors have addressed the comments raised in my previous review very well.

One very small change:

On line 238, I believe the word "liable" should actually be "labile".

Response: Point accepted. We have changed it in the revised manuscript.

The authors thank this reviewer for her/his time and constructive comments.